# LLMs Are Too Smart to Be Average: Controlling LLM Proficiency with Guided Decoding

## Abstract

Large Language Models (LLMs) are increasingly being used to simulate human behavior in applications such as educational technology, user modeling, and human-AI interaction. However, LLMs often default to expert-level reasoning, even when prompted to simulate individuals with limited or average proficiency. This misalignment limits their ability to realistically simulate users with diverse proficiency. In this work, we propose **Guided Decoding**, a decoding-time method for controlling the reasoning proficiency of LLMs during inference. Our approach fuses token-level logits from two sources: a reference prompt that elicits expert-level reasoning, and one or more guidance prompts that induce suboptimal reasoning patterns. By adjusting the contribution of these signals, our Guided Decoding enables fine-grained control over the model's reasoning behavior. Experiments on multiple question answering benchmarks demonstrate that our method not only modulates accuracy, but also influences the reasoning process, enabling more faithful simulation of users across a spectrum of proficiency levels.

## 1 Introduction

Large language models (LLMs) (Jiang et al., 2023a; Touvron et al., 2023; Achiam et al., 2023; Bai et al., 2023) have demonstrated remarkable capabilities across various role-conditioned tasks, such as student simulation (Zhang et al., 2024; Xu et al., 2024; Liu et al., 2024; Schmucker et al., 2024) and user simulation (Davidson et al., 2023; Zhang et al., 2025; Shao et al., 2023; Wu et al., 2025). In these tasks, LLMs consistently solve problems with high proficiency, often producing expert-level solutions regardless of the role they are instructed to simulate. However, this default to idealized behavior presents a limitation that LLMs are too smart to simulate individuals with average or limited proficiency (Benedetto et al., 2024). For example, when prompted to answer a question as an "average student", models frequently produce complete and correct solutions, failing to exhibit the mistakes, hesitations, or misconceptions that characterize real-world learners.

This lack of controllability limits the usefulness of LLMs in scenarios requiring realistic human-like simulation, such as educational assessment (Benedetto et al., 2024; Lu & Wang, 2024), personalized tutoring (Gao et al., 2025; Tang et al., 2025), psychological measurement (Yang et al., 2024) and healthcare (Li et al., 2024a). Although recent work like role-playing via in-context learning (Kong et al., 2024; Gao et al., 2025; Xu et al., 2024; Wang et al., 2025a) or post-training (Shao et al., 2023; Yu et al., 2024; Wang et al., 2024a) attempt to control the behavior of LLMs, these approaches often require detailed and extensive persona descriptions or interaction histories, which are impractical to scale across a large population. Broader demographic role-play offers greater scalability but lacks the resolution to distinguish reasoning proficiency within roles. Consequently, existing methods fall short in supporting **controllable proficiency simulation**, which refers to the ability to effectively model reasoning proficiency within LLMs, especially for simulating average individuals.

To address this gap, we suggest that, while prompting remains a powerful and flexible tool for steering LLM behavior, it often lacks the precision to modulate internal reasoning quality in a consistent and scalable way. Recent work has shown that decoding-time interventions can influence the reasoning trajectory (Li et al., 2022; Wang et al., 2025b), for example, by adjusting token probabilities to encourage exploration or mimic underthinking. These findings raise a natural research question: *can we simulate individuals with limited or average proficiency not only by changing the instruction prompts, but by shaping how it reasons during generation?*

To this end, we propose *Guided Decoding*, an effective method for simulating individuals with various reasoning proficiency levels during inference. Specifically, it guides the LLM generation process by fusing token-level logits from two sources: (1) a reference prompt that elicits expert-level reasoning, and (2) one or more guidance prompts that reflect flawed or misleading reasoning. By interpolating between these signals at each decoding step, our method enables fine-grained control over the model's reasoning proficiency. Unlike prior approaches, Guided Decoding does not require model fine-tuning with large-scale datasets, making it lightweight, interpretable, and broadly applicable.

Our contributions are summarized as follows: 1) To the best of our knowledge, we firstly formulate the **controllable proficiency simulation** problem, focusing on simulating individuals with varying proficiency levels. 2) We propose **Guided Decoding**, a novel decoding-time approach for user simulation. 3) Extensive experiments demonstrate that Guided Decoding successfully aligns reasoning capabilites and proficiency levels.

## 2 RELATED WORK

**Role-Playing with Language Models.** Role-playing is a paradigm for aligning language models with specific behaviors, tasks, or user expectations. Prior work (Chen et al., 2024) has established several types of role-playing settings. Demographic role-playing (Jiang et al., 2023b; Hong et al., 2024; Kong et al., 2024; LI et al., 2024) focuses on simulating a demographic of people, such as age, gender, and occupation. Character role-playing (Shao et al., 2023; Li et al., 2024b; Shaikh et al., 2025), on the other hand, simulates specific individuals, such as fictional or historical personas and preset individuals. From a technical standpoint, role-playing is typically implemented via in-context learning, using instructions to change LLMs' behavior (Wang et al., 2024a; Dai et al., 2025). Some works incorporate agent architectures or memory modules (Gao et al., 2025; Xu et al., 2024; Hong et al., 2024; Wang et al., 2024b) to sustain consistency in roles over long interactions. Others adopt parametric approaches, fine-tuning the model with domain-specific data to internalize persona characteristics (Shao et al., 2023; Yu et al., 2024; Wang et al., 2024a). While these methods effectively control style or behavior, they generally assume static or consistent personas and do not aim to control the underlying reasoning ability or proficiency level. In contrast, our work introduces a novel controllable proficiency simulation setting. Instead of assigning a character identity, we aim to simulate individuals with varying cognitive abilities. Technically, our method differs from prior work by controlling proficiency at decoding level, rather than prompt engineering or model fine-tuning. This enables fine-grained, dynamical, and model-agnostic control over reasoning behavior.

**LLM Decoding.** Decoding plays a key role in shaping the output behavior of LLMs. Some basic decoding methods (e.g., greedy (Germann, 2003) and beam search (Freitag & Al-Onaizan, 2017)) select tokens with the highest logits at each step. These methods perform well in many goal-driven tasks but often suffer from repetitive outputs. Additionally, there is growing interest in decoding methods that improve not only the fluency but also reasoning capability of LLM. *Contrastive decoding* (Li et al., 2022; O'Brien & Lewis, 2023; Phan et al., 2024; Chia et al., 2023) improves that by contrasting the outputs of a strong expert model with a weak amateur model. In parallel, decoding-time penalty mechanisms (Su & Collier, 2023; Wang et al., 2025b; Lightman et al., 2024; Heineman et al., 2024) have been proposed to alter the generation process. Repetition penalties (Su & Collier, 2023) discourage redundant token sequences, while Thought Switching Penalty (TIP) (Wang et al., 2025b) are designed to improve reasoning depth by lowering the logits of *thought-switching* tokens, encouraging deeper thinking. However, to our best knowledge, no prior work explicitly targets the controllable proficiency simulation problem through decoding. While existing methods are non-Monotonic and lack of fine-grained and probabilistic control, our work fills this gap by proposing a decoding-time strategy that enables fine-grained control over LLM reasoning ability.

## 3 PRELIMINARY

### 3.1 OBSERVATION OF EXPERT DEFAULT BIAS

Large language models exhibit a systematic tendency to produce high-proficiency responses regardless of role-conditioned instructions, which is a phenomenon we term *expert default bias*. We observe this pattern both qualitatively through individual examples and quantitatively across systematic

| You are an {**Role**} student. Please answer the question: Weng earns \$12 an hour for babysitting. Yesterday, she just did 50 minutes of babysitting. How much did she earn? | |
| --- | --- |
| **Role** | **Model Response** |
| **A-score** | To solve this, we first convert 50 minutes into hours: $\frac{50}{60} = \frac{5}{6}$. Then, $\frac{5}{6} \times 12 = 10$. Answer: Weng earned **\$10.00** for 50 minutes of babysitting. |
| **B-score** | Weng babysat for 50 minutes. That's $50 \div 60 = 0.833$ hours. $12 \times 0.833 \approx 10$. So she earned about **\$10**. |
| **C-score** | Weng earns \$12 an hour. She worked for 50 minutes, which is close to an hour. Maybe she earned around \$10. I'll just say **\$10**. |
| **D-score** | Weng earns \$12 an hour. She worked 50 minutes, so maybe like \$10? I think she earned **\$10**. |

Table 1: An example of GPT-4o responses under different role prompts.

experiments. Table 1 illustrates this phenomenon using GPT-4o prompted to simulate students of varying proficiency levels, with academic role descriptors ranging from A-score (high-performing) to D-score (low-performing). These scores are intended to loosely correspond to standard grading tiers, where A-score reflects expert-level reasoning and D-score represents struggling students. While surface tone and confidence differ across roles, the underlying reasoning and numerical outcome remain largely unchanged. The response differs only in superficial markers of uncertainty (e.g., "maybe like \$10?"), while still executing the correct calculation pathway. This observation is not an isolated case, but representative of a systematic pattern validated across our comprehensive experiments. As shown in Figure 3 and Table 2, prompting-based methods consistently maintain high accuracy across intended proficiency levels with minimal controllable variation, indicating poor alignment with target proficiency levels and irregular performance transitions.

### 3.2 PROBLEM STATEMENT

Let $\mathcal{P}$ denote the space of prompts (e.g., QA inputs), and let $\mathcal{Y}$ denote the space of possible outputs. We introduce a discrete set of proficiency levels $\mathcal{L} = \{0, 1, \ldots, N\}$, where $L \in \mathcal{L}$ represents the desired reasoning proficiency level, increasing from novice to expert. We define a generation function $f(L, P)$ that takes as input a prompt $P \in \mathcal{P}$ and a target proficiency level $L \in \mathcal{L}$, and outputs a response $O = f(L, P) \in \mathcal{Y}$ that simulates behavior at the specified level.

In cognitive modeling and educational assessment, psychometric frameworks such as Item Response Theory (IRT) and the Test Characteristic Curve (TCC) suggest that expected performance increases approximately linearly with latent ability, particularly in the mid-range of the scale (Embretson & Reise, 2013; Andrich, 2011). In parallel, proficiency levels are commonly defined by discretizing task performance into ordinal bands based on correctness or mastery rates (Beaton & Allen, 1992). This approach is widely adopted in AI for education, where simulated student profiles are constructed to represent varying levels of reasoning ability. Recent work (Lu & Wang, 2024; Benedetto et al., 2024) demonstrates that large language models can exhibit quasi-linear improvements in accuracy as proficiency increases. Motivated by these observations, we aim to construct $f$ such that the following properties are satisfied:

**Property 1: Local Consistency (Slope Deviation Score, SDS).** To ensure consistent transitions between adjacent proficiency levels, we define the empirical slope at level $L \in \{1, \ldots, N\}$ as:

$$\Delta_L = \mathbb{E}_{P \sim \mathcal{P}}[\mathcal{Q}(f(L, P)) - \mathcal{Q}(f(L-1, P))], \tag{1}$$

where $\mathcal{Q}(\cdot)$ denotes task-specific evaluation metrics (e.g., accuracy, correctness, or reasoning quality). Let the ideal linear slope be defined as:

$$\alpha = \frac{\mathbb{E}_{P \sim \mathcal{P}}[\mathcal{Q}(f(N, P))] - \mathbb{E}_{P \sim \mathcal{P}}[\mathcal{Q}(f(0, P))]}{N} \tag{2}$$

This ideal slope $\alpha$ represents the expected average performance gain per proficiency level under a perfectly linear progression from the lowest to the highest proficiency. The *Slope Deviation Score (SDS)* then measures the total deviation from this ideal slope across all transitions:

$$\text{SDS} = \sum_{L=1}^{N} |\Delta_L - \alpha| \tag{3}$$

Figure 1: Overview of our Guided Decoding framework for simulating reasoning proficiency in LLMs. A reference prompt elicits expert-level reasoning, simulating an expert user, while a set of guidance prompts introduce flawed or limited reasoning patterns to simulate amateur users. At each decoding step, the token logits from the expert stream are guided by those from the amateur stream via a fusion process controlled by a parameter $\lambda$. This guided decoding enables smooth interpolation between high- and low-proficiency responses, allowing fine-grained control over reasoning ability without modifying the underlying model.

A low SDS implies that the model's performance increases (or decreases) consistently across levels, preserving a smooth and predictable progression.

**Property 2: Global Alignment (Proficiency Deviation Score, PDS).** Let the ideal performance at level $L$ be defined as a linear interpolation between the endpoints:

$$\hat{\mathcal{Q}}(L) = \alpha \cdot L + \mathcal{Q}(f(0, P)), \tag{4}$$

We define the *Proficiency Deviation Score (PDS)* as the total deviation between actual and ideal performance across all intermediate levels:

$$\text{PDS} = \sum_{L=1}^{N-1} \left| \mathbb{E}_{P \sim \mathcal{P}}[\mathcal{Q}(f(L, P))] - \hat{\mathcal{Q}}(L) \right| \tag{5}$$

A lower PDS indicates that the model follows the expected proficiency–performance trajectory, avoiding reversals or irregular deviations across levels.

**Property 3: Interpretable Behavior Transitions.** The transition between outputs $f(L_1, P)$ and $f(L_2, P)$ for $L_1 < L_2$ should correspond to human-like improvements in reasoning, such as fewer logical errors and increased accuracy, rather than random or stylistic variation.

## 4 METHOD

As shown in Figure 1, to simulate users at different reasoning proficiency levels, we propose **Guided Decoding**, a decoding-time strategy that guides the expert with suboptimal reasoning streams from guidance prompts. This allows us to generate responses that reflect a target level of cognitive ability without modifying the underlying language model. Our method consists of two components, including Prompts Construction and Guided Decoding.

### 4.1 PROMPTS CONSTRUCTION

To simulate varying levels of reasoning proficiency, we construct two types of prompts: a reference prompt that elicits expert-level reasoning and a set of guidance prompts that emulate flawed reasoning patterns. An example of each type is shown in Figure 2. More details can be found in Appendix J.

**Reference Prompt.** As illustrated in the green box of Figure 2, the reference prompt is designed to encourage accurate, coherent, and logically complete reasoning. It typically includes explicit instructions and examples to steer the model toward expert-like behavior, such as that of a well-trained student or teacher. These prompts are carefully formulated to elicit correct computations, structured explanations, and precise conclusions. In our Guided Decoding framework, this reference prompt serves as the anchor for high-proficiency behavior.

**Guidance Prompts.** The guidance prompts are designed to mirror the systematic mistakes commonly made by humans. Unlike random perturbations, these prompts (i) are *task-aware*, allowing them to

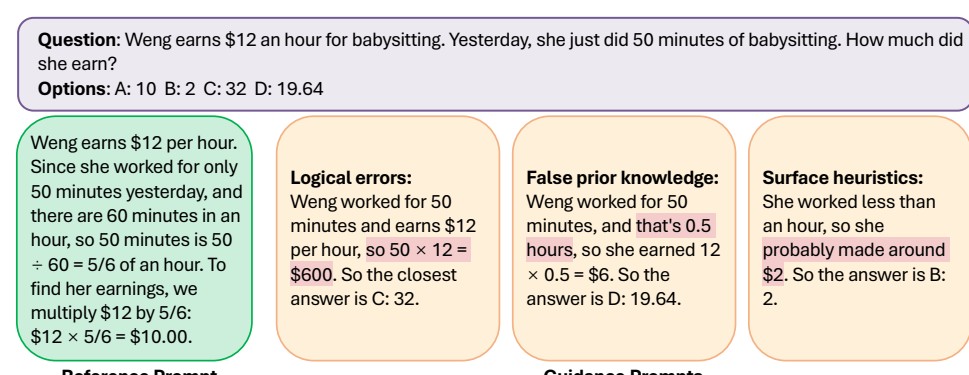

Figure 2: An illustration of Prompts Construction

be tailored to the reasoning demands of each domain, and (ii) are *interpretable*, as they correspond to cognitive pitfalls and thus provide meaningful control signals for proficiency simulation. In practice, we supply GPT with example questions and ask it to generate typical task-specific error patterns and corresponding example response, which capture realistic reasoning flaws. For instance, the guidance prompts, shown in the orange boxes of Figure 2, are designed to simulate reasoning errors commonly observed in average or struggling individuals for logic problems. By sampling from this pool, we obtain diverse yet structured suboptimal guidance signals that enable the LLM to emulate amateur behavior. These signals form the foundation for interpolation in our guided decoding framework.

## 4.2 GUIDED DECODING

Inspired by the success that decoding can adjust model reasoning process (Wang et al., 2025b; O'Brien & Lewis, 2023), we propose a novel decoding framework that modulates the model's output by fusing logits from both expert and amateur reasoning paths.

**Standard Decoding.** In original autoregressive decoding, a language model generates text by predicting one token at a time. At each position $p$, the model outputs a logit vector $\mathbf{z}_p \in \mathbb{R}^{|\mathcal{T}|}$, where $|\mathcal{T}|$ is the vocabulary size. The probability of generating a token $v \in \mathcal{T}$ is computed via the softmax function:

$$P(x_p = v \mid x_{<p}) = \frac{\exp(\mathbf{z}_{p,v})}{\sum_{v' \in \mathcal{T}} \exp(\mathbf{z}_{p,v'})} \tag{6}$$

where $\mathbf{z}_{p,v}$ is the token logit for token v. This decoding process relies entirely on a single prompt and the model's default reasoning trajectory. As a result, it inherently reflects the most likely or confident output of the model, which often corresponds to high-proficiency, expert-like reasoning—regardless of the role or ability level the user intends to simulate.

This design presents a key limitation for applications that require controllable simulation of reasoning proficiency. Standard decoding lacks the flexibility to adjust the depth, correctness, or fallibility of the model's reasoning once the prompt is fixed.

**Guided Decoding via Logit Fusion.** To address the aforementioned issue, we introduce our Guided Decoding framework that introduces an auxiliary control mechanism that modulates the token-level logits using multiple prompts.

***Reference Logits*** ($\mathbf{z}_p^{\text{ref}}$). The reference logits are obtained by prompting the LLM with a carefully crafted reference prompt $P_{\text{ref}}$ aimed at eliciting expert-level reasoning. To mitigate the *expert-default bias*, we flatten this signal by applying a down-weighting factor: $\mathbf{z}_p^{\text{ref}} = \beta * P(x_p = v \mid x_{<p})$, where $\beta \in (0, 1]$ is a hyperparameter that reduces the model's inherent tendency to default toward correct answers, thereby enabling more balanced interpolation with the flawed guidance logits.

***Guidance Logits*** ($\mathbf{z}_p^{\mathbf{g}}$). The guidance logits are computed from a set of $K$ guidance prompts $\{P_g^k\}_{k=1}^K$, each crafted to elicit flawed or average-level reasoning. For each $P_g^k$, we obtain the token logits $\mathbf{z}_p^k$ at position $p$. Rather than average weighting, we adopt similarity-aware aggregation that aggregates each guidance logit's contribution based on its divergence from the reference reasoning path.

We first compute a distance-like score between each guidance logit vector and the reference logits using the complement of cosine similarity:

$$s^k = 1 - \cos\left(\mathbf{z}_p^k, \mathbf{z}_p^{\text{ref}}\right) \tag{7}$$

A lower value of $s^k$ indicates a higher similarity to the reference. To convert these distances into aggregation weights, we apply a average over the negated scores, and the final guidance logits are then computed as a weighted sum:

$$\mathbf{z}_p^{\text{g}} = \sum_{k=1}^{K} \alpha^k \cdot \mathbf{z}_p^k, \text{ where } \alpha^k = \frac{s^k}{\sum_{j=1}^{K}(s^j)} \tag{8}$$

This similarity-based fusion ensures that guidance prompts that are more deviated from the expert reasoning path exert greater influence. Intuitively, when a flawed reasoning stream diverges from the expert logits, its lower similarity score increases its contribution to the fused logits. This design allows the model to intentionally incorporate the characteristic errors encoded in those flawed streams when simulating lower proficiency, enabling controlled degradation in reasoning.

***Logits Fusion***. The final fused logits $\mathbf{z}_p$ used for token prediction are computed by interpolating between the reference and guidance logits:

$$\mathbf{z}_p = \lambda \cdot \mathbf{z}_p^{\text{ref}} + (1 - \lambda) \cdot \mathbf{z}_p^{\text{g}}, \tag{9}$$

The fusion weights are normalized such that their sum is 1, enabling smooth interpolation between the two logit sources. $\lambda$ is the fusion parameter that corresponds to the proficiency level. To relate $\lambda$ to discrete proficiency levels, we define it using a level index $l \in \{1, 2, \ldots, N\}$ as: $\lambda = l/N$, where $N$ denotes the total number of simulated proficiency levels. A higher $l$ value corresponds to higher proficiency. In particular, $l = N$ leads to minimal influence from guidance and simulates a high-performing user. Conversely, $l = 1$ corresponds to the lowest proficiency level, where the model is maximally influenced by flawed guidance.

This mapping ensures that $\lambda$ increases monotonically with decreasing proficiency, enabling controlled degradation of reasoning behavior. As $\lambda$ decreases, the model becomes increasingly susceptible to distractive cues encoded in the guidance prompts. Two extreme settings help illustrate the behavior of the fusion mechanism. When $\lambda = 0$, the fusion reduces to $\mathbf{z}_p = \mathbf{z}_p^{\text{g}}$, and the model generates responses entirely under the influence of the guidance prompt, reflecting the most error-prone reasoning behavior. On the other hand, when $\mathbf{z}_p = \mathbf{z}_p^{\text{ref}}$, it recovers the expert response. This setting is useful as a reference point for full-proficiency generation.

## 5 EXPERIMENTS

### 5.1 EXPERIMENT SETTINGS

**Datasets.** To evaluate the simulation ability of different levels of average individuals, we adopt student simulation in line with prior research (Lu & Wang, 2024; Benedetto et al., 2024). We experiment on four datasets. The **RACE** dataset (Lai et al., 2017) is a multiple-choice question answering benchmark derived from English reading comprehension exams. The **GSM8K** dataset (Cobbe et al., 2021) is a fundamental mathematical question-answering dataset that require multi-step reasoning for resolution. The **StrategyQA** dataset (Geva et al., 2021) is a question-answering benchmark focusing on open-domain commonsense questions. To assess generalization ability, we adopt multi-turn scenario simulations to model the *politeness* of human interaction. The **Mutual** dataset (Cui et al., 2020) is a multi-turn dialogue benchmark evaluating multi-turn conversations with reasoning-intensive response selection. More details can be found in Appendix A.

**Baselines.** To evaluate the effectiveness of our Guided Decoding framework, we compare against four representative baselines, covering prompt engineering, role-play conditioning, and student simulation approaches. Specifically, we include **zero-shot prompting** (Kojima et al., 2022), and **few-shot prompting** (Wei et al., 2022). These approaches represent standard prompting strategies. **betterRP** (Kong et al., 2024) improves role-playing by injecting task-aware persona descriptions into the prompt. **studentSIM** (Benedetto et al., 2024) designs a student simulation method by involving the estimation of question difficulty, which helps simulation alignment for different level of student.

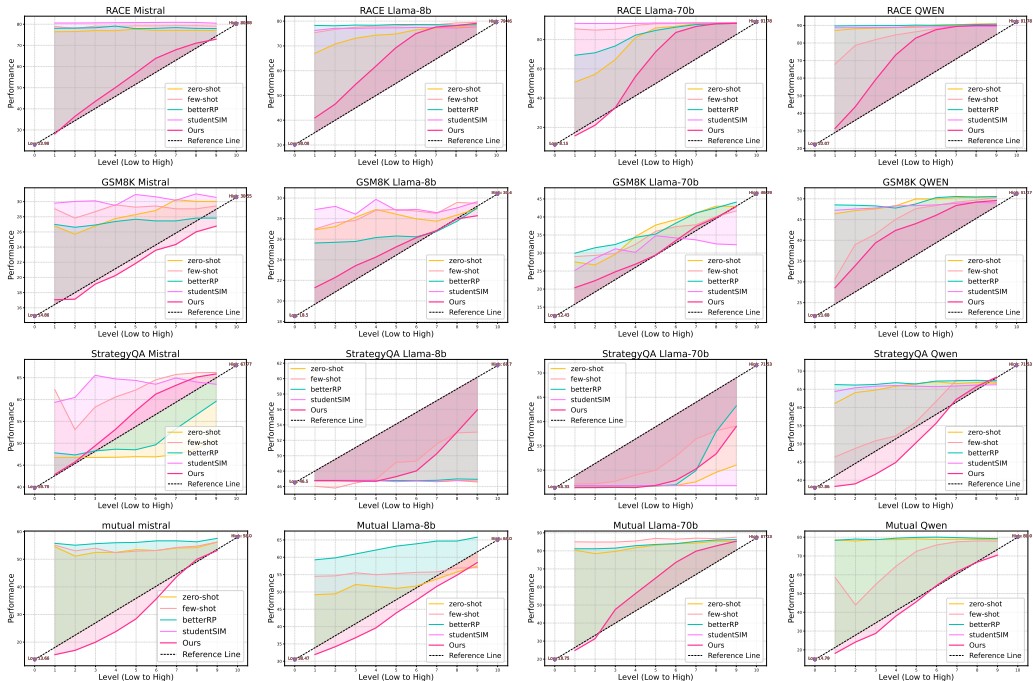

Figure 3: Main results across four datasets and four model backbones. Each curve represents performance across nine simulated average proficiency levels. The lowest level corresponds to the pure poor simulation, defined as the lowest performance achieved when simulating poor individuals under each method. The highest level corresponds to the pure expert simulation. The reference line connects these two endpoints. The vertical deviation between each method's curve and the reference line reflects the Proficiency Deviation Score (PDS), while the angular deviation from the reference line indicates the Slope Deviation Score (SDS), capturing local inconsistency across levels.

**Metrics.** To evaluate the simulation performance, we use accuracy to reflect how well the model performs under different simulated proficiency settings and two designed metrics introduced in Section 3.2, *Proficiency Deviation Score (PDS)* and *Slope Deviation Score (SDS)*. (1) **Proficiency Deviation Score (PDS)**, as shown in Eq. equation 5, quantifies the global alignment between the actual proficiency–performance curve and the ideal linear trajectory from pure novice to pure expert. It penalizes reversals or inconsistent deviations in expected performance, and is particularly sensitive to large gaps or misordered trends across intermediate levels. (2) **Slope Deviation Score (SDS)**, as shown in Eq. equation 3, captures the local consistency of reasoning transitions by measuring how closely the empirical slope between adjacent levels aligns with the ideal uniform slope $\alpha$. It reflects the smoothness and monotonicity of simulated proficiency shifts.

## 5.2 MAIN RESULTS

The results across four datasets and four backbone models are presented in Figure 3 and Table 2. We assess each method's ability to simulate reasoning proficiency through nine average levels.

**Global Alignment: Proficiency Deviation Score (PDS).** Across most settings, Guided Decoding achieves the lowest PDS, reducing PDS by tens to hundreds of points. For example, on RACE with Mistral, our method achieves a PDS of **30.78**, compared to as high as **262.62** from the studentSIM baseline, resulting a approximately $7\times$ lower. This substantial improvement highlights the effectiveness of Guided Decoding in aligning outputs with the desired proficiency trend.

In some cases, such as on StrategyQA with LLaMA models, we observe elevated PDS across all methods. This is likely due to hallucination issues in the base model, leading to uniformly low performance and large deviations from the ideal linear progression. These results suggest that while Guided Decoding remains effective, the underlying model quality and dataset characteristics still influence the extent of controllability.

| Dataset | Methods | Mistral | | Llama-8b | | Llama-70b | | Qwen | |
|---|---|---|---|---|---|---|---|---|---|
| | | PDS↓ | SDS↓ | PDS↓ | SDS↓ | PDS↓ | SDS↓ | PDS↓ | SDS↓ |
| RACE | zero-shot | 229.20 | 45.05 | 176.47 | 28.43 | 254.64 | **43.34** | 291.77 | 52.07 |
| | few-shot | 247.54 | 45.55 | 208.46 | 35.32 | 295.82 | 62.66 | 245.05 | 40.71 |
| | betterRP | 240.40 | 45.75 | 212.90 | 39.22 | 295.83 | 45.18 | 298.72 | 54.73 |
| | studentSIM | 262.62 | 45.74 | 201.70 | 38.10 | 370.72 | 66.62 | 289.21 | 55.07 |
| | Ours | **30.78** | **15.24** | **89.44** | **19.92** | **112.65** | 49.79 | **134.49** | **44.26** |
| GSM8K | zero-shot | 50.01 | 9.29 | 33.06 | 7.47 | 57.91 | 14.85 | 112.62 | 19.50 |
| | few-shot | 56.93 | 12.17 | 36.36 | 7.02 | 51.46 | 14.97 | 73.52 | 15.80 |
| | betterRP | 44.07 | 11.72 | 20.33 | 6.31 | 64.70 | 13.00 | 115.71 | 21.55 |
| | studentSIM | 68.35 | 11.79 | 41.29 | 9.26 | 58.42 | 22.51 | 106.89 | 21.84 |
| | Ours | **9.56** | **4.26** | **7.68** | **2.54** | **12.84** | **6.96** | **53.52** | **12.49** |
| StrategyQA | zero-shot | 67.25 | 18.38 | 66.20 | 12.34 | 101.45 | 15.88 | 100.23 | 21.44 |
| | few-shot | 74.97 | 23.16 | 45.67 | 9.89 | **62.84** | **11.00** | **27.70** | 16.87 |
| | betterRP | 38.94 | 13.72 | 65.54 | 11.99 | 78.44 | 21.06 | 110.84 | 25.89 |
| | studentSIM | 89.37 | 22.77 | 66.19 | 12.25 | 109.08 | 20.16 | 103.16 | 25.11 |
| | Ours | **19.99** | **8.11** | **45.38** | **9.69** | 87.28 | 14.96 | 30.14 | **11.70** |
| Mutual | zero-shot | 159.24 | 33.77 | 57.10 | 19.38 | 261.13 | 49.05 | 281.69 | 52.06 |
| | few-shot | 163.31 | 34.34 | 82.13 | 25.13 | 293.99 | 51.31 | 176.96 | 50.73 |
| | betterRP | 183.28 | 33.67 | 134.73 | 21.08 | 270.60 | 48.82 | 287.34 | 51.27 |
| | Ours | **36.97** | **14.68** | **30.63** | **4.63** | **72.45** | **23.92** | **21.90** | **13.44** |

Table 2: Quantitative results across four datasets and four model backbones. Lower PDS/SDS indicates closer alignment with target proficiency and smoother degradation.

**Local Consistency: Slope Deviation Score (SDS).** In addition to global alignment, Guided Decoding consistently achieves the lowest SDS across most settings, indicating its ability to produce smooth and monotonic performance transitions across adjacent proficiency levels. In many cases, our method reduces SDS by more than half compared to other baselines. For example, on Mutual with LLaMA-8b, Guided Decoding achieves an SDS of **4.63**, while other methods range from 19.38 to 25.13, showing that our method is over $4\times$ more consistent in stepwise behavior. Notably, even on challenging datasets like RACE, where the reasoning process is complex and multi-faceted, Guided Decoding maintains strong local consistency. On Mistral and LLaMA-8b, SDS is reduced by over 50% compared to the best-performing baseline, demonstrating the effectiveness of logit-level interpolation in avoiding erratic changes or reversals between levels. In contrast, zero-shot and few-shot prompting exhibit frequent fluctuations in SDS due to their lack of structured progression. Similarly, role-play prompting fails to enforce consistent behavior across levels, as its reliance on high-level role descriptions lacks the granularity required for fine-tuned control.

**Visualization Insights.** The proficiency–performance curves provide qualitative evidence for the effectiveness of Guided Decoding. Unlike baseline methods, whose performance curves frequently exhibit non-monotonic trends, Guided Decoding consistently produces smooth and monotonic trajectories that align with the target proficiency levels. Specifically, we observe violations of monotonicity, particularly with studentSIM and BetterRP on GSM8K and RACE, where intermediate proficiency levels sometimes outperform both lower and higher levels. These reversals are symptomatic of uncontrolled reasoning dynamics, where prompts fail to precisely modulate the model's internal generation process. Furthermore, in most cases, model performance exceeds the reference line across levels. This supports our central hypothesis that LLMs are often "too smart to be average," tending to default to expert-like reasoning regardless of the intended role or assigned proficiency level.

## 5.3 QUALITATIVE RESULTS

As shown in Table 3, Guided Decoding produces qualitatively distinct reasoning patterns across simulated proficiency levels. Specifically, The expert response offers a fully correct and well-structured solution, performing intermediate calculations, and clearly identifying the savings. This response serves as the high-proficiency anchor. At the average Level 3, the model produces a numerically correct answer but compresses the reasoning into a single arithmetic operation. While the final answer is correct, it skips the comparison of total costs and lacks an explanation of why each step is valid. This reflects a *surface heuristic*, a mild reasoning shortcut in Guidance signals. At

| **Question:** Vincent can buy flowers in packages of 3 for \$2.50 or in packages of 2 for \$1. How much money does he save by buying 18 flowers at the better price? | |
|---|---|
| **Proficiency** | **Model Response** |
| **Expert** | Vincent can buy flowers in packages of 3 for \$2.50 or in packages of 2 for \$1. So, if he buys 18 flowers, he can buy 6 packages of 3 flowers for \$2.50 each, or 9 packages of 2 flowers for \$1 each. The total cost of 6 packages of 3 flowers is $6 \times \$2.50 = \$15.00$. The total cost of 9 packages of 2 flowers is $9 \times \$1 = \$9.00$. So, Vincent saves \$15.00 - \$9.00 = **\$6.00** by buying 18 flowers at the better price. |
| **Average (6)** | Vincent can buy flowers in packages of 3 for \$2.50 or in packages of 2 for \$1. If he buys 18 flowers, he can buy 6 packages of 3 flowers for \$2.50 each, or 9 packages of 2 flowers for \$1 each. So, he saves $\$2.50 \times 6 - \$1 \times 9 = $ **\$6.00**. |
| **Average (3)** | Vincent can buy flowers in packages of 3 for \$2.50 or in packages of 2 for \$1. So, if he buys 18 flowers, he will buy 6 packages of 3 flowers and 2 packages of 2 flowers. The total cost is $6 \times 2.50 + 2 \times 1 = $ **\$15.00**. |
| **Poor (1)** | Vincent can buy flowers in packages of 3 for \$2.50 or in packages of 2 for \$1. He buys 18 flowers at the better price. He saves $\$2.50 \times 18 = $ **\$45.00**. |

Table 3: Model responses at different simulated proficiency levels for a math problem.

| # Guidance | Performance at Simulated Levels (1 to 9) | | | | | | | | | PDS | SDS |
|---|---|---|---|---|---|---|---|---|---|---|---|
| few-shot | 78.97 | 77.99 | 79.11 | 79.03 | 79.25 | 79.24 | 79.31 | 79.31 | 79.10 | 247.54 | 45.55 |
| 1 | 68.45 | 79.04 | 78.55 | 79.31 | 79.52 | 79.45 | 79.45 | 79.38 | 79.59 | 238.97 | 44.30 |
| 2 | 38.92 | 51.39 | 63.23 | 71.23 | 75.90 | 78.41 | 79.59 | 79.80 | 79.87 | 154.57 | 35.09 |
| 3 | 28.13 | 36.42 | 43.45 | 50.00 | 56.82 | 63.85 | 67.90 | 71.03 | 72.91 | **30.78** | **15.24** |

Table 4: Ablation study on the number of guidance prompts.

average Level 6, the issue becomes more severe as the reasoning becomes structurally flawed. The model incorrectly combines two packaging types to reach 18 flowers, violating the constraint that all flowers must be bought from a single option. This reflects a *logical error*. Additionally, at the poor Level 9, the model's reasoning deteriorates further, exhibiting compounded flaws drawn from all types of guidance errors. Such errors are indicative of low proficiency reasoning, where even basic problem facts are misunderstood or misapplied. Overall, these examples demonstrate that Guided Decoding produces not just varying answers, but also reasoning shifts in the reasoning process.

## 5.4 ABLATION STUDY

To examine how the number of guidance prompts influences controllability, we vary the number of guidance prompts used in Guided Decoding from 1 to 3 and measure performance across simulated proficiency levels on the RACE dataset, as shown in Table 4. The few-shot baseline demonstrates little variation across simulated levels. In contrast, even with a single guidance prompt, our method produces a modest performance gradient, yielding slightly lower PDS and SDS. Adding a second guidance prompt significantly improves simulation performance. With three guidance prompts, the model achieves the best results, exhibiting smoother transitions and a clearer distinction across simulated proficiency levels. This highlights that introducing more diverse guidance prompts enhances the model to simulate a more realistic range of intermediate reasoning states.

## 6 CONCLUSION

In this work, we formulate the problem of *controllable proficiency simulation* in large language models. To address that, we propose *Guided Decoding*, a novel decoding-time approach for simulating varying levels of reasoning proficiency in large language models. By interpolating between reference and guidance signals during inference, our method enables fine-grained control over simulation proficiency without requiring model fine-tuning. Experimental results across multiple datasets and models demonstrate that Guided Decoding more accurately reflects intended proficiency levels compared to existing prompting and role-play baselines. This work highlights the potential of decoding-time control as a scalable solution for cognitively grounded LLM simulation.

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

# A DATASET

To cover a comprehensive range of subject for student simulation. We experiment on four datasets. The **StrategyQA** dataset (Geva et al., 2021) is a question-answering benchmark focusing on open-domain questions, requiring implicit reasoning to infer the necessary steps from the question itself through a strategic approach. It is designed to evaluate the ability to perform implicit reasoning, necessary for answering questions that do not have direct or explicit answers within the text.

The **GSM8K** dataset (Cobbe et al., 2021) is structured to facilitate question answering on fundamental mathematical problems that require multi-step reasoning for resolution. The solutions to these problems primarily involve performing a sequence of elementary calculations using basic arithmetic operations.

The **RACE** dataset (Lai et al., 2017) is a multiple-choice question answering benchmark derived from English reading comprehension exams. It tests the ability to comprehend complex passages, extract relevant information, and answer questions that require inferential and contextual understanding, making it ideal for simulating reading-based academic tasks.

The **MuTual** dataset (Cui et al., 2020) is a multiple-choice benchmark for multi-turn dialogue reasoning, constructed from English listening comprehension exams. Each instance presents a dialogue context with subtle distractors and requires selecting the most appropriate next utterance from several candidates. Unlike single-turn QA datasets, MuTual emphasizes pragmatic understanding, dialogue coherence, and reasoning over conversational flow, making it a challenging testbed for simulating interactive reasoning and communication proficiency.

# B IMPLEMENTATION DETAILS

In our experiments, we simulate 11 discrete proficiency levels corresponding to scores from 0 to 100 in increments of 10 (i.e., {0, 10, 20, ..., 100}). We evaluate our method using four backbone models: Mistral-7B (Jiang et al., 2023a), LLaMA3-8B, and LLaMA3-70B (Touvron et al., 2023) and Qwen3 (Yang et al., 2025). All reference and guidance answers are generated by the GPT-4o model. The questions are taken from the corresponding training set. We set $\beta$ as 0.5 empirically. All generations are produced using greedy decoding. Experiments are conducted on a combination of NVIDIA RTX A5000 GPUs (24GB) and NVIDIA A100 GPUs (80GB), depending on model size.

# C ABLATION ON EXPERT-DEFAULT BIAS TEMPERATURE ($\beta$)

To investigate the impact of mitigating the expert-default bias, we conduct an ablation study on the down-weighting factor $\beta$ using the RACE dataset with Mistral as the backbone. Table 5 summarizes the results across different $\beta$ values. We find that moderate settings (e.g., $\beta = 0.5$) provide the best trade-off, yielding the lowest PDS (30.78) and a competitive SDS (15.24). In contrast, very small or large $\beta$ values either diminish the corrective influence of the reference logits or excessively flatten the expert signal, both of which harm performance. Nevertheless, our method consistently achieves lower PDS and SDS compared to other baselines, validating the robustness of our approach.

| $\beta$ | 0.3 | 0.4 | 0.5 | 0.6 | 0.7 |
|---|---|---|---|---|---|
| PDS↓ | 74.96 | 34.93 | **30.78** | 65.93 | 107.51 |
| SDS↓ | 17.12 | 13.23 | **15.24** | 16.96 | 20.67 |

Table 5: Ablation study on the expert-default bias factor $\beta$. Moderate values (e.g., $\beta = 0.5$) yield the best balance between PDS and SDS.

# D EFFECT OF SIMILARITY-BASED LOGIT SCALING

We also ablate the impact of similarity-based scaling when aggregating guidance logits. In our default setting, we weight each guidance prompt based on its cosine similarity to the reference logits at each decoding step. This weighting ensures that guidance signals more aligned with the expert trajectory

| Model | Sim | Performance at Levels 1–9 | | | | | | | | | PDS | SDS |
|---|---|---|---|---|---|---|---|---|---|---|---|---|
| Mistral | N | 34.47 | 40.53 | 49.51 | 57.31 | 66.50 | 72.84 | 76.32 | 78.69 | 79.59 | 91.99 | 20.20 |
| | Y | 28.13 | 36.42 | 43.45 | 50.00 | 56.82 | 63.85 | 67.90 | 71.03 | 72.91 | **30.78** | **15.24** |
| LLaMA-8b | N | 42.86 | 48.46 | 55.36 | 62.74 | 69.84 | 75.13 | 77.71 | 78.20 | 78.69 | 96.06 | **18.83** |
| | Y | 40.95 | 46.45 | 54.31 | 61.63 | 69.15 | 74.93 | 77.64 | 78.20 | 79.11 | **89.44** | 19.92 |
| LLaMA-70b | N | 14.97 | 22.21 | 34.54 | 56.54 | 72.49 | 85.58 | 89.48 | 91.22 | 91.43 | 117.19 | 50.27 |
| | Y | 14.28 | 21.52 | 33.42 | 54.73 | 71.58 | 84.82 | 88.92 | 90.80 | 91.09 | **112.65** | **49.78** |

Table 6: Ablation study on the effect of similarity-based logit scaling on guidance logits.

have greater influence during fusion. We compare this to a variant that uses uniform averaging over guidance logits without similarity scaling.

As shown in Table 6, similarity-based scaling yields consistent improvements in both Proficiency Deviation Score (PDS) and Slope Deviation Score (SDS) across all model sizes on the RACE dataset. For example, on Mistral, similarity-based scaling reduces PDS from **91.99** to **30.78** and SDS from **20.20** to **15.24**. Similar improvements are observed on LLaMA-8b (PDS: **96.06** → **89.44**) and LLaMA-70b (PDS: **117.19** → **112.65**).

These results suggest that similarity scaling enhances the quality of fusion by attenuating noisy or overly divergent guidance signals. The gains are most significant on smaller models like Mistral, where reasoning is more sensitive to input variation. On stronger models like LLaMA-70b, the gains are more modest, possibly because the model is already capable of filtering incoherent signals internally.

Overall, this ablation validates the role of similarity-aware fusion in improving both global alignment and local consistency in proficiency simulation.

# E    REASONING DIVERSITY EVALUATION

To directly assess the variety of reasoning errors generated by our method, we conducted a targeted evaluation using LLM-as-a-Judge (Zheng et al., 2023). We randomly sampled 100 incorrect responses produced on the GSM8K dataset. Each response was then analyzed using GPT-4o to classify the dominant error type(s) according to our predefined taxonomy. The distribution of error types was as follows: 44% of responses contained logical errors, 22% reflected false prior knowledge, and 34% were attributed to surface heuristics. Additionally, we asked GPT-4o to assess whether multiple error types occurred within a single response. The analysis revealed that 19% of the responses exhibited two or more error types, indicating that our method can simulate compound or mixed reasoning failures.

We also provide a qualitative example: *Question: Raymond and Samantha are cousins. Raymond was born 6 years before Samantha. Raymond had a son at the age of 23. If Samantha is now 31, how many years ago was Raymond's son born?*

*Model Answer: Raymond was born 6 years before Samantha. So Raymond is 6 years older than Samantha. Raymond had a son at the age of 23. So Raymond's son is 23 years old. Samantha is now 31. So Raymond's son is 31 - 23 = 8 years old.*

This response exhibits both a logic error, incorrectly aligning the timeline of Raymond's age with Samantha's, and a surface heuristic, namely the use of direct subtraction between salient numbers without temporal justification.

# F    QUANTITATIVE REASONING-QUALITY ANALYSIS

We randomly sampled 100 questions from GSM8K and generated responses at five different proficiency levels. We then employed GPT-4o as an LLM-as-a-Judge to systematically identify and categorize reasoning errors in the generated chain-of-thought (CoT) responses. As shown in Tab. 7,

| Proficiency level (low to high) | 1 | 3 | 5 | 7 | 9 |
|---|---|---|---|---|---|
| Logical errors | 115 | 125 | 109 | 104 | 95 |
| False prior knowledge | 9 | 7 | 5 | 3 | 4 |
| Surface-heuristic mistakes | 72 | 60 | 57 | 47 | 47 |
| **Avg. errors per request** | **1.96** | **1.92** | **1.71** | **1.54** | **1.46** |

Table 7: Quantitative reasoning quality analysis

we observe that across all error categories, the number of reasoning mistakes decreases monotonically as proficiency increases. The average errors per request drop from 2.00 to 1.58 between low and high proficiency. This shows that Guided Decoding affects not only answer correctness but also the internal structure of reasoning, producing progressively fewer logical mistakes, heuristic shortcuts, and incorrect assumptions at higher proficiency levels.

# G    ADDITIONAL EXPERIMENTS ON LARGER BACKBONES

To examine if our method work on larger models, we conduct additional experiments on STRATE-GYQA using larger variants of the Mistral and Qwen families. Specifically, we evaluate Mistral–8B/24B and Qwen–8B/14B/32B under four settings: Zero-shot, Few-shot, BetterRP, and our Guided Decoding method. The results are shown in Table 8.

| Model | Method | PDS ↓ | SDS ↓ |
|---|---|---|---|
| **Mistral 8B** | Zero-shot | 67.25 | 18.38 |
| | Few-shot | 74.97 | 23.16 |
| | BetterRP | 38.94 | 13.72 |
| | **Ours** | **19.99** | **8.11** |
| **Mistral 24B** | Zero-shot | 71.03 | 31.65 |
| | Few-shot | 78.68 | 44.82 |
| | BetterRP | **48.35** | 34.17 |
| | **Ours** | 65.75 | **28.65** |
| **Qwen 8B** | Zero-shot | 100.23 | 21.44 |
| | Few-shot | **27.70** | 16.87 |
| | BetterRP | 110.84 | 25.89 |
| | **Ours** | 30.14 | **11.70** |
| **Qwen 14B** | Zero-shot | 60.80 | 16.79 |
| | Few-shot | 98.59 | 14.77 |
| | BetterRP | 118.79 | 19.66 |
| | **Ours** | **25.99** | **16.26** |
| **Qwen 32B** | Zero-shot | 44.19 | 18.87 |
| | Few-shot | 72.82 | 31.04 |
| | BetterRP | 75.36 | 26.96 |
| | **Ours** | **23.11** | **12.36** |

Table 8: Results on STRATEGYQA using larger Mistral and Qwen variants. Lower PDS and SDS indicate closer imitation of the target proficiency level.

From the results, we notice that there is **No systematic reversal with scale.** Across both model families, increasing the backbone size (e.g., Qwen–14B/32B, Mistral–24B) does not lead to few-shot prompting outperforming our method. Guided Decoding remains highly competitive, often achieving the best PDS and SDS scores even as the backbone grows significantly. In addition, our method achieves **Consistent advantage.** Our method consistently yields the lowest or near-lowest PDS and SDS across all evaluated settings. This suggests that Guided Decoding scales reliably with model capacity and does not suffer from degradation when applied to larger LLMs.

# H  ABLATION STUDY: NUMBER OF GUIDANCE PROMPTS

In the main paper, we reported that using three guidance prompts produced the strongest performance *among the evaluated settings*. Our intention was not to claim that three prompts is globally optimal, but rather that it was the best choice within the tested configuration. To further clarify this point, we extend the ablation to include four and five guidance prompts. The complete results are presented in Table 9.

| # Prompts | PDS ↓ | SDS ↓ |
|---|---|---|
| 1 | 238.97 | 44.30 |
| 2 | 154.57 | 35.09 |
| 3 | **30.78** | **15.24** |
| 4 | 37.81 | 15.45 |
| 5 | 37.96 | 15.24 |

Table 9: Ablation on the number of guidance prompts. Lower PDS and SDS indicate better proficiency alignment.

The results reaffirm that using three guidance prompts is a strong and efficient default choice. While increasing the number of prompts to four or five yields comparable performance, the improvements over three prompts are marginal. Additionally, for resource-constrained scenarios, even two prompts provide substantial gains relative to one prompt or zero guidance baseline methods. Overall, three prompts offer an effective balance between controllability, stability, and computational cost.

# I  INFERENCE OVERHEAD JUSTIFICATION

While our method requires computing logits from both reference and guidance prompts, theoretical **inference overhead is negligible**, since all forward passes can be processed in a single batched call. That is, we concatenate the reference and guidance prompts into a batch and run a single forward pass through the LLM decoder at each step. This allows efficient parallelization on GPU. However, we acknowledge that this batching increases GPU memory usage linearly with the number of guidance prompts.

# J  PROMPTS

```
Prompt Template for RACE

You will be shown a multiple choice question from an English
reading comprehension exam. You must assign a difficulty level
to the given multiple choice question,and select the answer
choice that a student would pick. Here are some examples.

## Example 1 ##

{example_question}

You must assign a difficulty level to the given multiple
choice question,and select the answer choice that a **good**
level student would pick.

Please first think about the reasoning process that a **good**
level student would follow to select the answer,including
the misconceptions that might cause them to make mistakes.
```

And then give me the answer. Format: Reasoning process.
<Answer>your choice</Answer>

**Your Answer:**

{example_response_good}

## Example 2 ##

{example_question}

You must assign a difficulty level to the given multiple
choice question,and select the answer choice that a **poor**
level student would pick.

Please first think about the reasoning process that a **poor**
level student would follow to select the answer,including
the misconceptions that might cause them to make mistakes.
And then give me the answer. Format: Reasoning process.
<Answer>your choice</Answer>

**Your Answer:**

{example_response_bad}

## Example End ##

Reading passage: "{context}"

Question: "{question}"

Options: "{option}"

You must assign a difficulty level to the given multiple
choice question,and select the answer choice that a
**{student_level}** level student would pick.

Please first think about the reasoning process that a
**{student_level}** level student would follow to select the
answer,including the misconceptions that might cause them to
make mistakes. And then give me the answer. Format: Reasoning
process. <Answer>your choice</Answer>

**Your Answer:**

---

**RACE Example Question**

Reading passage: "People who are about the same age as you are
your peers. Peers include your friends and your classmates.
They have strong influence on your actions. Your peers
influence how you think,how you act,and even how you dress.
Peer pressure is the influence that people of the same age
have on one another.

---

Sometimes your peers influence you in a helpful or positive
way. A friend may notice that you have problems in math. And
he might invite you to join a study group or show you how
to solve a difficult problem during lunch. Such actions are
helpful to you.
Sometimes your peers influence you in a  or unhealthy way. A
friend might offer you cigarettes .Cigarettes are harmful to
your health. Your friend knows that. Your friend also knows
that underage smoking is against the law. Yet he or she still
makes the offer. This bad influence is negative peer pressure.
Your peers may not always realize they are influencing you
in a negative way. For example. a friend might invite you
to the movies. You would love to go,but you have a lot of
housework to do. In situations like this you should make a
wise decision.
You can learn to deal with negative peer pressure. Keep away
from people who try to influence your behavior in harmful
ways. Though it is not always easy to say no,itś really
necessary to learn to do that. Follow the following steps.
First. look at the person and say no in a firm but polite
voice. Make sure your face shows that you are serious. Let
the person know you will not back down. Then,give reasons
for saying no. Explain why you wonť do what the person wants.
Remember to say goodbye and walk away if he or she continues."
Question: "Which sentence shows the writerś opinion?"
Options: "A: Peer pressure is the influence that people of the
same age have on one another.
B: We should learn to deal with negative peer pressure.
C: Itś not always important to say no to your peers.
D: Peers include your friends and your classmates.
"

**RACE Reference Example Answer**

<Answer>B</Answer>
Reasoning process:
The student would recognize that the question is asking
for the writer's opinion,which is different from a factual
statement. They would analyze each option and identify which
one contains a subjective perspective or a recommendation.
Option A is a factual definition of peer pressure,and options
C and D also present factual or descriptive statements.
However,option B includes the word "should," which indicates
advice or guidance. This suggests that the writer is sharing
their opinion on how to handle negative peer pressure.
Therefore,a good student would select option B as the correct
answer.answer.

**RACE Guidance Example Answers**

<Answer>A</Answer>
Reasoning process:

The student might struggle to distinguish between facts and opinions. They might think that since the sentence provides a clear and straightforward definition of peer pressure,it must be the writer's opinion. They could focus on the literal meaning of the sentence rather than evaluating whether it expresses a subjective viewpoint or advice. The student may overlook the importance of words like "should" in option B,which actually signals the writer's opinion,and instead choose a more neutral or general statement that seems important in the passage.

<Answer>C</Answer>
Reasoning process:
The student might misunderstand the passage due to unfamiliar vocabulary like "peer pressure," "influence," or "negative." They may pick option C because the phrase "not always important" sounds like advice or something the writer could be saying. Without fully understanding the context or meaning of the sentence,they might guess based on the tone of the words,not realizing that option C actually contradicts the writer's main message. Limited vocabulary comprehension leads the student to focus on surface-level language rather than meaning.

<Answer>D</Answer>
Reasoning process:
The student might simply choose the first sentence they remember from the passage without thinking about whether it shows an opinion. They may confuse something that *sounds like it came from the writer* with something that *is the writer's opinion*. Since the sentence in option D talks about who peers are,the student might think,"Well,the writer said that,so it must be their opinion." This reflects a misunderstanding of the difference between stating a fact and expressing a viewpoint. The student is likely relying on memory rather than critical reading.

---

**StrategyQA Example Question**

Question: "Is a Boeing 737 cost covered by Wonder Woman (2017 film) box office receipts?" Options: "A: True B: False"

---

**StrategyQA Reference Example Answer**

<Answer>A</Answer>
Reasoning process:
The average cost of a US Boeing 737 plane is 1.6 million dollars. Wonder Woman (2017 film) grossed over 800 million dollars at the box office. So,the Answer is A.

**StrategyQA Guidance Example Answer**

<Answer>B</Answer>
Reasoning process:
Making a movie costs a lot of money. Since Wonder Woman probably spent millions to make,there's no way the movie made enough profit to cover a plane like a Boeing 737. So the answer is B.

<Answer>B</Answer>
Reasoning process:
A Boeing 737 costs over a billion dollars,and movies don't make that much,so Wonder Woman couldn't cover the cost. Therefore,the answer is B.

<Answer>B</Answer>
Reasoning process:
Wonder Woman is a movie character,and she doesn't buy airplanes. So she would not cover the cost of a Boeing 737. Therefore,the answer is B.

---

**Mutual Guidance Example Question**

f : hi ,jack . could you help me carry these books ? m : i 'd love to ,mary ,but i need to meet professor johnson in his office immediately . Options: A: "f : you are going to your own office immediately .",B: "f : you will help me carry the books first . thank you .",C: "f : you are going to help me carry these books before you go to professor johnson 's office immediately . you are so nice .",D: "f : you are going to professor johnson immediately ."""

---

**Mutual Reference Example Answer**

<Answer>D</Answer>
Reasoning process:
The female speaker (f) should acknowledge his reason for declining. Option D is a simple,logical paraphrase of his statement,showing understanding without contradicting him. So,the Answer is D.

---

**Mutual Guidance Example Answer**

e<Answer>C</Answer>
Reasoning process:
The female speaker (f) doesn't care if Jack has somewhere to be,he can help me first. It's rude of him to just walk away when I'm clearly struggling.
So,the Answer is C.

## K    LIMITATIONS

While Guided Decoding enables controllable simulation of reasoning proficiency without model fine-tuning, it comes with several limitations. First, our method is designed for open-source models, where the decoding process can be customized; so applying it to closed-source models is challenging due to restricted access to internal decoding mechanisms. Second, although Guided Decoding can steer the model's reasoning behavior, it is inherently constrained by the reasoning patterns provided in the guidance prompts. As a result, it cannot generate entirely novel reasoning trajectories beyond those guidance signals.

## L    USE OF LARGE LANGUAGE MODELS

We used large language models solely for grammar checking and minor language refinement. No parts of the content were fully generated by an LLM.

