# OpenReview forum: "LLMs Are Too Smart to Be Average: Controlling LLM Proficiency via Guided Decoding"
_ICLR.cc/2026/Conference — Submitted to ICLR 2026_

### Official Review · Reviewer_oYbY · 2025-10-15

**Soundness:** 2
**Presentation:** 2
**Contribution:** 2
**Rating:** 2
**Confidence:** 5

**Summary:**

The paper proposes a method for controlling the proficiency level of a LLM when answering questions. By providing reference and guidance prompts, the authors instruct the LLM to generate both professional and intentionally flawed responses. During decoding, they interpolate between the logits produced under these two prompting conditions to control the proficiency level of the final answer.

**Strengths:**

* Does not require fine-tuning.
* Model agnostic approach.
* Can be applied in a plug and play manner.

**Weaknesses:**

* The paper is difficult to read and lacks clarity in presentation.
* It does not include a clear measurement or evaluation of reasoning proficiency levels when controlling the proficiency level by their method.
* There is insufficient evidence to support the claimed bias in proficiency levels (only one example is shown).
* Contains several verbal errors.

**Questions:**

* How do you conclude that "the model continues to apply high-proficiency reasoning regardless of the intended role"? In particular, you state "Even the D-score response converges on the correct answer of $10, revealing that the model continues to apply high-proficiency reasoning regardless of the intended role" Please justify this conclusion.

* How did you observe an "expert default bias"? Is this claim based on a single example, or on multiple cases? Please provide quantitative evidence (counts, proportions, or statistical tests) to support the existence of this bias.

* Are the statements in the last paragraphs of Section 4.2 (the text after Eq. 8) correct? Specifically:
  * "This similarity-based fusion ensures that guidance prompts more aligned with the expert reasoning path are weighted more heavily"
  * "A lower lvalue corresponds to higher proficiency. In particular, l= N leads to minimal influence from guidance and simulates a high-performing user."
  * "As λ increases, the model becomes increasingly susceptible to distractive cues encoded in the guidance prompts."


* Do you have quantitative results showing that controlling the proficiency level of answers also affects reasoning quality? In the paper you report only final answer quality. Please report metrics that capture intermediate reasoning quality (e.g., chain of thought correctness or other reasoning diagnostics) when proficiency is varied.

* On the StrategyQA benchmark, you report that with a smaller Llama variant your method outperforms alternatives, while with a larger variant few-shot prompting is consistently better. Do you have an intuition for why model size reverses the relative performance? Can you run experiments with larger versions of the other two LLMs to test whether this effect is due to model size rather than model family?

---

> ### Author Response · Authors · 2025-11-19
> **Response by Authors to Reviewer oYbY(1/2)**
>
> We sincerely thank the reviewer for the time and effort dedicated to evaluating our paper. We address the concerns point by point below:
>
> > W1: Clarity and verbal errors.
>
> A: We thank the reviewer for the careful reading. We have revised the manuscript accordingly.
>
> > W2, Q1, Q2: How do you observe an "expert default bias"? Please provide quantitative evidence.
>
> A:
> Thank you for this important question. We clarify that our conclusion about the "expert default bias" is supported by both systematic quantitative evidence and illustrative qualitative examples.
>
> 1. **Quantitative Evidence from Baseline Experiments**: The primary evidence for expert default bias comes from our systematic evaluation of prompting-based baselines across all datasets. As shown in Figure 3 and Table 2. Zero-shot and few-shot prompting consistently produce **high accuracy across all intended proficiency levels**, with minimal variation despite explicit role instructions. Across all four datasets and four model backbones (16 experimental settings), prompting-based methods consistently show high PDS and SDS scores. This systematic pattern is what we term the "expert default bias", where LLMs tend to apply near-optimal reasoning strategies regardless of role instructions.
>
> 2. **Table 1 as an Illustrative Example**: Table 1 serves as a concrete illustration of this phenomenon, not the sole basis for our claim. The example demonstrates that when GPT-4o is prompted to simulate different student proficiency levels (A-score through D-score), all responses converge on the correct answer with similar reasoning structures. Critically, even the D-score response still executes the correct calculation, differing only in **superficial acting** (e.g., "maybe like 10?"). This demonstrates that despite the explicit request to behave as a weak solver, the model maintains high-quality reasoning and produces correct solutions. This example is representative of the broader pattern observed across our preliminary experiments and validated quantitatively in our main results (Figure 3, Table 2), where role-conditioned prompts consistently failed to elicit genuine reasoning quality and performance degradation.
>
> > Q3: Statement correction.
>
> A: Thank you for the careful reading. We apologize for the confusion caused by the unclear phrasing in the original text and have revised the statements to better reflect our design intent. The mathematical formulation in Equations (7)–(9) remains correct.
>
> Specifically, we have clarified the similarity-based weighting mechanism as follows:
>
> *This similarity-based fusion ensures that guidance prompts that are more deviated from the expert reasoning path exert greater influence. Intuitively, when a flawed reasoning stream diverges from the expert logits, its lower similarity score increases its contribution to the fused logits. This design allows the model to intentionally incorporate the characteristic errors encoded in those flawed streams when simulating lower proficiency, enabling controlled degradation in reasoning.*
>
> Our key design principle is that, by emphasizing divergent guidance signals, we ensure that the fused guidance logits represent genuinely degraded reasoning patterns rather than minor variations of expert behavior, which is essential for controllable proficiency simulation.
>
> We have also revised the clarification to make sure it aligns with the formula.

---

> ### Author Response · Authors · 2025-11-19
> **Response by Authors to Reviewer oYbY(2/2)**
>
> > W3, Q4: How does controlling proficiency affect reasoning quality, and can this be shown quantitatively?
>
> A:
> Thank you for raising this important point. We agree that reasoning-quality diagnostics complement final-answer accuracy. We have conducted additional quantitative analysis to address this concern.
>
> **Quantitative reasoning-quality analysis (GSM8K)**
> We randomly sampled 100 questions from GSM8K and generated responses at five different proficiency levels. We then employed GPT-4o as an LLM-as-a-Judge to systematically identify and categorize reasoning errors in the generated chain-of-thought (CoT) responses.
> Following the error taxonomy established in our guidance prompt design for GSM8k (Section 4.1), we instructed the judge model to identify three major types of reasoning errors: Logical errors, False prior knowledge, and Surface-heuristic mistakes
>
> The results are summarized below:
>
> | Proficiency level (low to high)| 1 | 3 | 5 | 7 | 9 |
> |------------------------|---|---|---|---|---|
> | Logical errors | 115 | 125 | 109 | 104 | 95 |
> | False prior knowledge | 9 | 7 | 5 | 3 | 4 |
> | Surface-heuristic mistakes | 72 | 60 | 57 | 47 | 47 |
> | **Avg. errors per request**| **1.96** | **1.92** | **1.71** | **1.54** | **1.46** |
>
> From the results, we observe that across all error categories, the number of reasoning mistakes **decreases monotonically** as proficiency increases. The average errors per request drop from **2.00** to **1.58** between low and high proficiency. This shows that Guided Decoding affects not only answer correctness but also the **internal structure of reasoning**, producing progressively fewer logical mistakes, heuristic shortcuts, and incorrect assumptions at higher proficiency levels.
>
>
> > Q5: What explains the size-dependent performance reversal on StrategyQA, and is it caused by model size or model family?
>
> A:
> Thank you for the insightful question. We address both parts of the question below.
>
> 1. **Intuition for the Llama-size effect.**
>    As discussed in Section 5.2 (lines 374-377), we think that the elevated PDS across all methods on StrategyQA with LLaMA models is primarily attributable to hallucination issues inherent to the base model, which lead to uniformly low performance and large deviations from the ideal linear proficiency progression.
>
>    Critically, as shown in Figure 3, this pattern is not unique to our method—all baseline methods (zero-shot, few-shot, BetterRP, studentSIM) exhibit similar non-monotonic and irregular trajectories on StrategyQA with LLaMA models. This suggests that the issue stems from the base model's difficulty with the dataset rather than a specific failure mode of Guided Decoding.
>
> 2. **Additional experiments with larger Qwen and Mistral backbones.**
>    To test whether the observed effect is due to model size versus model family, we conducted additional experiments on StrategyQA using larger variants of Qwen and Mistral. The results are shown below:
>
> | Model | Method | PDS ↓ | SDS ↓ |
> |-------|--------|--------|--------|
> | **Mistral 8B** | Zero-shot | 67.25  | 18.38 |
> |               | Few-shot  | 74.97  | 23.16 |
> |               | BetterRP  | 38.94  | 13.72 |
> |               | **Ours**      | **19.99**  | **8.11** |
> | **Mistral 24B** | Zero-shot | 71.03  | 31.65 |
> |                | Few-shot  | 78.68  | 44.82 |
> |                | BetterRP  | **48.35**  | 34.17 |
> |                | **Ours**      | 65.75  | **28.65** |
> | **Qwen 8B** | Zero-shot | 100.23 | 21.44 |
> |            | Few-shot  | **27.70**  | 16.87 |
> |            | BetterRP  | 110.84 | 25.89 |
> |            | **Ours**      | 30.14  | **11.70** |
> | **Qwen 14B** | Zero-shot | 60.80  | 16.79 |
> |             | Few-shot  | 98.59  | 14.77 |
> |             | BetterRP  | 118.79 | 19.66 |
> |             | **Ours**      | **25.99**  | **16.26** |
> | **Qwen 32B** | Zero-shot | 44.19  | 18.87 |
> |             | Few-shot  | 72.82  | 31.04 |
> |             | BetterRP  | 75.36  | 26.96 |
> |             | **Ours**      | **23.11**  | **12.36** |
>
> We notice that:
> - **No systematic reversal with scale:** Across both Mistral and Qwen families, scaling to larger models (14B, 24B, 32B) does not cause few-shot prompting to outperform Guided Decoding. In fact, our method maintains consistently lower PDS/SDS scores in most cases.
>
> - **Consistent advantage**: Across most experiments, Guided Decoding consistently achieves the lowest or near-lowest PDS and SDS, demonstrating robust performance scaling.

---

> ### Author Response · Authors · 2025-11-27
>
> Dear Reviewer oYbY,
>
> Thank you very much for the time and effort you’ve devoted to reviewing our paper. We have carefully addressed the concerns you raised and provided detailed explanations in our rebuttal and revised the manuscript. If any part of our response remains unclear or would benefit from further clarification, we would be grateful for the opportunity to elaborate. Thank you!
>
> Authors

---

### Official Review · Reviewer_wkof · 2025-10-28

**Soundness:** 3
**Presentation:** 3
**Contribution:** 3
**Rating:** 6
**Confidence:** 3

**Summary:**

The paper "LLMs Are Too Smart to Be Average: Controlling LLM Proficiency via Guided Decoding" proposes Guided Decoding, a novel approach to modulate the proficiency level of answers generated by large language models (LLMs). The method first uses GPT-4o to produce a reference prompt and several guidance prompts for a given question, then extracts logit vectors from a white-box LLM. Through a specially designed fusion mechanism, the model can generate responses at different proficiency levels, simulating answers from students with varying capabilities. Experiments show that Guided Decoding achieves the best or near-best scores on both SDS and PDS metrics across multiple QA tasks compared to all baselines.

**Strengths:**

- The paper proposes an innovative token-level decoding framework that effectively simulates student responses at different proficiency levels. This has meaningful implications for AI-driven education and adaptive tutoring systems.
- The paper is well-written and easy to follow. The experimental results are convincing within the presented scope and demonstrate clear improvements over prior work.

**Weaknesses:**

- There are a few minor typos — for example: Line 82: missing a period. Line 262: incorrect notation for the reference prompt.

- The experiments are somewhat limited. While the paper reports SDS and PDS scores and includes a basic ablation study, it lacks further empirical analysis. For instance, the paper mentions that using 3 guidance prompts yields the best performance, but no experiments on more guidance prompts to support this. Furthermore, the computation cost and inference time needs to be discussed.

- Some notation and consistency issues affect readability. In Section 3.2, $L$ is introduced as the proficiency level, whereas in Section 4.2, $l$ is used with an opposite interpretation (ascending vs. descending proficiency). Section 5.3 then extends the latter usage. It would be clearer to unify the notation and interpretation throughout the paper.

**Questions:**

1. The effectiveness of Guided Decoding may depend on the capability of the base LLM. For example, if the base model performs poorly on a given dataset, Guided Decoding might not be able to produce a full range of responses (e.g., A+ to F) but only within a narrower band (e.g., around C). How do you plan to address this limitation or adapt the method to weaker base models?

2. In Section 3.2, Property 3 appears to be somewhat subjective. Could the authors clarify how this property could be quantitatively evaluated or verified?

3. The method operates via token-level sequential modifications. However, token-level differences do not always correspond to meaningful semantic differences at the sentence or paragraph level. For example, two answers may differ only in the order of explanation (reason-first vs. answer-first) but convey equivalent understanding. How does the proposed method handle or account for this?

4. In line 264, the paper states that $\beta$ reduces the model’s inherent tendency to default toward correct answers. Could the authors explain why this adjustment is necessary, given that $\lambda$ in Equation (9) already controls the bias toward higher proficiency responses?

---

> ### Author Response · Authors · 2025-11-19
> **Response by Authors to Reviewer wkof(1/2)**
>
> We sincerely thank the reviewer for the time and effort dedicated to evaluating our paper. We address the concerns point by point below:
>
> > W1: There are a few minor typos.
>
> A: We thank the reviewer for the careful reading, and we have revised the manuscript accordingly.
>
> > W2: Additional ablation study on more guidance prompts and discuss the the computation cost and inference overhead.
>
> A:
> Thank you for this helpful suggestion. We have expanded our empirical analysis in two directions.
>
> 1. **Ablation with more guidance prompts.**
>    Originally, we stated that using 3 guidance prompts yielded the strongest performance *among the settings we evaluated*. To avoid overclaiming, we did not intend to assert that 3 prompts is globally optimal. To address your concern, we conducted additional experiments using 4 and 5 guidance prompts. The full results are shown below:
>
>    | #Prompts | PDS ↓ | SDS ↓ |
>    |----------|--------|--------|
>    | 1 | 238.97 | 44.30 |
>    | 2 | 154.57 | 35.09 |
>    | 3 | **30.78** | **15.24** |
>    | 4 | 37.81 | 15.45 |
>    | 5 | 37.96 | 15.24 |
>
>    These results support using 3 guidance prompts as a reasonable default, balancing controllability quality with computational efficiency. For resource-constrained scenarios, even 2 prompts provide substantial improvement over baseline methods. Using 4 or 5 prompts shows only marginal differences compared to 3 prompts.
>
> 2. **Computation cost and inference overhead.**
>    As discussed in Appendix F, our method requires computing logits from both the expert (reference) and guidance prompts. The **inference overhead is negligible in practice**, because all streams are processed in **one batched forward pass** through the decoder at each step. This parallelization allows efficient GPU execution without increasing latency. We acknowledge, however, that the batching increases **GPU memory usage linearly** with the number of guidance prompts. In typical settings (3–5 prompts), the memory cost remains manageable, and no additional model training or fine-tuning is required.
>
> > W3: It would be clearer to unify the notation and interpretation throughout the paper.
>
> A: Thank you for identifying this inconsistency. We apologize for the confusion, and we have revised it accordingly. Specifically, when $l$ is small, it means low proficiency, and when $l$ is larger, it corresponds to higher proficiency level. In the revised version, we have unified the notations and interpretations throughout the paper.
>
> > Q1: Guided Decoding may be limited by the underlying model’s capability. How can Guided Decoding handle cases where the base model is too weak to express the full range of proficiency levels?
>
> A: We thank the reviewer for this thoughtful question. Meanwhile, we would like to clarify our problem formulation and explain why the observed behavior aligns with the intended design of our method.
>
> 1. **Proficiency is Relative to Model Capability:**
> The key insight  of our framework is that proficiency levels are defined relative to the base model's capability range, not against an absolute external standard. In our framework:
>
>    - Highest proficiency (L = N): The best performance the model can achieve with expert-level reasoning (optimal prompting), corresponding to A+.
>    - Lowest proficiency (L = 0): The worst performance the model produces while still meaningfully engaging with the task (maximally degraded reasoning via guidance prompts), corresponding to F.
>
> 1. **When Model Capability Becomes a Practical Concern**. We acknowledge that model capability does impose constraints on the absolute range of proficiency that can be simulated, which is a **common fundamental challenge** shared by all simulation methods. For scenarios requiring simulation across very broad absolute performance ranges, practitioners could **select appropriately capable base models** for the target task, and **use multiple models for different proficiency bands**.

---

> ### Author Response · Authors · 2025-11-19
> **Response by Authors to Reviewer wkof(2/2)**
>
> > Q2: How can reasoning quality (Property 3) be quantitatively evaluated or validated?
>
> A: Thank you for raising this important point. We have conducted additional quantitative analysis to address this concern.
>
> **Quantitative reasoning-quality analysis (GSM8K)**
> We randomly sampled 100 questions from GSM8K and generated responses at five different proficiency levels. We then employed GPT-4o as an LLM-as-a-Judge to systematically identify and categorize reasoning errors in the generated chain-of-thought (CoT) responses.
> Following the error taxonomy established in our guidance prompt design for GSM8k (Section 4.1), we instructed the judge model to identify three major types of reasoning errors: Logical errors, False prior knowledge, and Surface-heuristic mistakes.
>
> The results are summarized below:
>
> | Proficiency level (low to high)| 1 | 3 | 5 | 7 | 9 |
> |------------------------|---|---|---|---|---|
> | Logical errors | 115 | 125 | 109 | 104 | 95 |
> | False prior knowledge | 9 | 7 | 5 | 3 | 4 |
> | Surface-heuristic mistakes | 72 | 60 | 57 | 47 | 47 |
> | **Avg. errors per request**| **1.96** | **1.92** | **1.71** | **1.54** | **1.46** |
>
> From the results, we observe that across all error categories, the number of reasoning mistakes **decreases monotonically** as proficiency increases. The average errors per request drop from **2.00** to **1.58** between low and high proficiency. This shows that Guided Decoding affects not only answer correctness but also the **internal structure of reasoning**, producing progressively fewer logical mistakes, heuristic shortcuts, and incorrect assumptions at higher proficiency levels. The new results and discussions have been added to the revised paper.
>
> > Q3: The method operates via token-level sequential modifications. However, token-level differences do not always correspond to meaningful semantic differences at the sentence or paragraph level. How does the proposed method handle or account for this?
>
> A: We thank the reviewer for this insightful observation. This is indeed a critical consideration for our method. We have specifically designed our approach to address this concern through two key mechanisms:
>
> 1. **Unified Response Format**: Both the reference (expert) and guidance (amateur) prompts are constructed to follow the same structural format, like reasoning process first, followed by the final answer. This design choice ensures that token-level differences between the expert and amateur streams reflect differences in reasoning content rather than mere stylistic or organizational variations.
> 2. **Synchronized Token Generation**: At each decoding step, once a token is selected through our logit fusion process (Eq. 9), this token is immediately appended to all active generation streams (both reference and guidance). This synchronization ensures that all prompts share the same generated history at each step, maintaining parallel reasoning trajectories that differ only in the quality of reasoning choices, not in structural divergence.
>
> Together, these mechanisms ensure that the token-level differences induced by our method correspond to meaningful cognitive distinctions rather than semantically equivalent reformulations.
>
> > Q4: What does β do, and why is β needed?
>
> A: Thank you for this insightful question about the design rationale for the parameter β. While both β and λ influence the balance between expert and amateur reasoning, they serve distinct and complementary roles in our framework.
>
> 1. **Distinct Roles of β and λ**:
>    - **λ (Equation 9)**: Controls the relative weighting between reference (expert) and guidance (amateur) logits during fusion. This is the primary proficiency control mechanism that users adjust to **simulate different ability levels and inject different reasoning processes**.
>    - **β** Applies a pre-fusion down-weighting to the reference logits before they enter the interpolation process. This addresses the **inherent** expert default bias of LLMs by flatting the distribution. Concretely, without β-based flattening, the expert logits​ may have very sharp probability distributions (high confidence in correct tokens), while the guidance logits have softer distributions (representing uncertainty in flawed reasoning). Therefore, we equalize the dynamic range of the two logit sources using β.
> 2. **Empirical Validation**: As shown in Table 5 in Appendix C, we conduct the ablation on expert-default bias temperature (beta). In the results, we found that very small or large β values either diminish the corrective influence of the reference logits or excessively flatten the expert signal, both of which harm performance.
>
> | β     | 0.3   | 0.4   | 0.5   | 0.6   | 0.7   |
> |-------|-------|-------|-------|-------|-------|
> | **PDS ↓** | 74.96 | 34.93 | **30.78** | 65.93 | 107.51 |
> | **SDS ↓** | 17.12 | 13.23 | **15.24** | 16.96 | 20.67  |

---

> ### Author Response · Authors · 2025-11-27
>
> Dear Reviewer wkof,
>
> Thank you very much for the time and effort you’ve devoted to reviewing our paper. We appreciate your positive feedback. We have carefully addressed the concerns you raised and provided detailed explanations in our rebuttal and revised the manuscript. If any part of our response remains unclear or would benefit from further clarification, we would be grateful for the opportunity to elaborate. Thank you!
>
> Authors

---

### Official Review · Reviewer_CaA5 · 2025-11-01

**Soundness:** 2
**Presentation:** 3
**Contribution:** 2
**Rating:** 4
**Confidence:** 3

**Summary:**

This paper addresses the "expert default bias" of LLMs, where models tend to produce high-proficiency answers even when prompted to simulate average or novice users. The authors formulate the problem of "controllable proficiency simulation" and propose Guided Decoding, a decoding-time method that controls reasoning proficiency. The method works by fusing token-level logits from a high-proficiency "reference prompt" with one or more low-proficiency "guidance prompts" that emulate flawed reasoning. The authors introduce two metrics, Proficiency Deviation Score and Slope Deviation Score, to evaluate the alignment and consistency of simulated proficiency. Experiments across four datasets and several models show the method's ability to modulate performance more smoothly than prompting-based baselines.

**Strengths:**

1. The paper clearly identifies and formulates a relevant and timely problem: the "expert default bias" of LLMs and the need for "controllable proficiency simulation". This is significant for applications in education, user modeling, and human-AI interaction.
2.The introduction of the PDS and SDS metrics provides a quantitative framework for evaluating this new problem, moving beyond simple accuracy to measure global alignment and local consistency.

**Weaknesses:**

1. The method's effectiveness hinges on the availability of high-quality "guidance prompts" that capture "realistic reasoning flaws". The paper suggests using GPT to generate these, but this process is not deeply explored. This reliance on generated or manually-crafted flawed examples seems difficult to scale and may introduce its own biases. The ablation study (Table 4) confirms that performance is highly dependent on the number of guidance prompts.
2. The core evaluation metrics (PDS and SDS) are predicated on an assumption that task performance increases linearly with latent ability. This assumption, while motivated by psychometric theory, is a strong and potentially inaccurate simplification. The validity of the primary quantitative results rests heavily on this untested assumption of linearity.
3. The core mechanism of interpolating logits from an "expert" and an "amateur" stream is a variation of existing techniques like contrastive decoding, which the paper cites. While the application to proficiency control is new, the technical novelty of the decoding strategy itself is limited.

**Questions:**

see weaknesses

---

> ### Author Response · Authors · 2025-11-19
> **Response by Authors to Reviewer CaA5(1/2)**
>
> We sincerely thank the reviewer for the time and effort dedicated to evaluating our paper. We address the concerns point by point below:
>
> > W1: Does the method depend too heavily on high-quality, potentially biased guidance prompts, and can this process scale?
>
> A:
> Thank you for raising the concern about the construction and scalability of guidance prompts. We address each aspect below.
>
> 1. **Guidance prompts represent generic cognitive-error patterns rather than handcrafted item-specific solutions.**
>    As described in Sec. 4.1, we provide GPT with a few representative example questions and ask it to generate *task-specific error patterns* that capture realistic reasoning flaws. These flawed reasoning chains reflect well-established categories of cognitive mistakes, such as premature deduction, surface heuristics, and faulty numerical conversions, rather than manually authored or item-level errors.
>
> 2. **The prompts are generated automatically with only a small, one-time design cost per task.**
>    GPT synthesizes the flawed reasoning chains directly from the seed examples, and the resulting small set of 3–5 guidance prompts is reused across all items. This avoids per-item engineering and yields a **lightweight, scalable workflow** compared to collecting or crafting human error traces.
>
> 3. **Bias risk is minimal because the prompts encode cognitive fallacies, not personas or socio-cultural attributes.**
>    The flawed reasoning patterns model *how* a solution goes wrong, not *who* is producing the solution. They do not encode demographic information or value-laden traits, and thus do not propagate demographic bias. They are inherently content-agnostic reasoning mistakes.
>
> 4. **The method intentionally models a controlled family of reasoning errors for the purpose of proficiency controllability.**
>    As noted in our Limitations section, the simulated reasoning space is bounded by the guidance patterns we specify. This is an intentional design choice: the goal is to produce **consistent, interpretable, and controllable** proficiency levels, rather than to exhaustively enumerate all possible incorrect reasoning chains. Additional reasoning styles can be incorporated simply by generating extra guidance prompts.
>
> 5. **The approach is naturally scalable because adding new error types requires no algorithmic changes.**
>    To adapt to a new task or domain, one simply provides GPT with a few seed questions to generate the relevant flawed reasoning patterns. No modification to the decoding mechanism is required, enabling seamless extension to additional reasoning styles or datasets.
>
> **Summary.**
> Our approach uses **automatically generated, task-aligned cognitive-error patterns**, maintains low bias risk, scales with minimal overhead. Empirically, all guidance prompts provide controllable proficiency modulation, consistent with our goal of modeling reasoning proficiency.

---

> ### Author Response · Authors · 2025-11-19
> **Response by Authors to Reviewer CaA5(2/2)**
>
> > W2: Is the assumed linear relationship between proficiency and performance valid?
>
> A: Thank you for raising this important methodological question. We address the concern from two aspects.
> - **Psychometric Theory**
>   We adopt a linear performance trajectory because it aligns with **psychometric conventions**. The **Test Characteristic Curve (TCC)**, which maps latent ability to expected score, is approximately linear in the **mid-range of ability**, where test items are concentrated and where our evaluation focuses [1,2,3]. This modeling assumption enables **consistent evaluation** without requiring manual calibration for every task or dataset.
>
> - **Educational Simulation Practice**
>   We define proficiency levels based on **final task performance**, where observed scores are discretized into ordinal proficiency bands for interpretability and reporting [4,5]. This approach is widely adopted in **AI for education**, where simulated student profiles are assigned based on **correctness rates or mastery levels**. Many studies assume or observe **quasi-linear improvements** in performance metrics [6,7,8,9].
>   For example:  Lu et al.[6] simulate increasing student proficiency levels that yield **monotonic accuracy gains**. Benedetto et al.[7] vary LLM student profiles and show **progressive reasoning improvements**.
>
> [1] Baker, F. B. *The Basics of Item Response Theory*. ERIC Clearinghouse, 2001.
> [2] Embretson, S. E., & Reise, S. P. *Item Response Theory for Psychologists*. Lawrence Erlbaum Associates, 2000.
> [3] Hambleton, R. K., Swaminathan, H., & Rogers, H. J. *Fundamentals of Item Response Theory*. Sage Publications, 1991.
> [4] Beaton, A. E., & Allen, N. L. (1992). *Interpreting scales through scale anchoring*. Journal of Educational Measurement.
> [5] OECD. *PISA 2018 Assessment and Analytical Framework*. OECD Publishing, 2019.
> [6] Lu, X., & Wang, X. (2024). *Generative Students: Using LLM-Simulated Student Profiles to Support Question Item Evaluation*. Proceedings of ACM Learning\@Scale.
> [7] Benedetto, L., et al. (2024). *Using LLMs to Simulate Students' Responses to Exam Questions*. EMNLP 2024.
> [8] Piech, C., et al. (2015). *Deep Knowledge Tracing*. NIPS.
> [9] Yeung, C.-K., Wang, Y., & Liu, Q. (2019). *Deep-IRT: Make Deep Learning Based Knowledge Tracing Explainable Using Item Response Theory*. arXiv:1904.11738.
>
> > W3: How technically novel is the proposed logit-interpolation mechanism beyond existing contrastive decoding methods?
>
> A: Thank you for this comment. We respectfully argue that our contribution extends significantly beyond existing contrastive decoding techniques in multiple fundamental ways:
>
> 1. **Novel Problem Formulation: Controllable Proficiency Simulation**
>    Prior contrastive decoding focuses on *improving* generation quality by contrasting strong and weak models to suppress undesirable patterns. In contrast, our goal is fundamentally different: we introduce the problem of **Controllable reasoning proficiency simulation**, where the model must generate responses across a spectrum of proficiency levels.
>
> 2. **Distinct Technical Mechanism and Design Rationale**
>    Contrastive decoding subtracts a weak model or weak prompt to sharpen reasoning. In contrast, we construct **multiple flawed reasoning processes**, each encoding a distinct cognitive error type derived from task examples. These streams provide **interpretable bases** for interpolation, enabling smooth and controllable proficiency modulation. The mechanism is designed to **simulate structured reasoning errors**, not to denoise a strong model. Our similarity-weighted fusion further adjusts guidance adaptively, which is absent from prior logit-combination methods.
>
> 3. **Novel Evaluation Framework**
>    Beyond the decoding rule, our contribution includes defining a **new task** and developing **new evaluation metrics** (PDS/SDS) that quantify global and local consistency across simulated proficiency levels. These components are essential to the problem and are not part of previous decoding frameworks.
>
> We also provide the table below for clearer reference.
> | Aspect | Contrastive Decoding | Guided Decoding (Ours) |
> |--------|---------------------|------------------------|
> | Logit operation | Subtraction | Interpolation |
> | Guidance sources | Single weak signal | Multiple task-aware error patterns |
> | Aggregation | N/A (single source) | Similarity-weighted fusion |
> | Controllability | Binary | Continuous  |

---

> ### Author Response · Authors · 2025-11-27
>
> Dear Reviewer CaA5,
>
> Thank you very much for the time and effort you’ve devoted to reviewing our paper. We have carefully addressed the concerns you raised and provided detailed explanations in our rebuttal and revised the manuscript. If any part of our response remains unclear or would benefit from further clarification, we would be grateful for the opportunity to elaborate. Thank you!
>
> Authors

---

### Author Response · Authors · 2025-12-02
**Summary of the Rebuttal**

Dear Area Chairs and Reviewers,

We sincerely thank you all for the constructive feedback and appreciation of our work. Below, we summarize strengths highlighted by reviewers and respond to major concerns, explicitly attributing each point.

## **Acknowledged Strengths**
We would like to briefly restate the core contributions of our paper.

First, we **define the new task of proficiency-controlled reasoning**, which Reviewers **CaA5** and **wkof** identify as a clearly articulated and underexplored problem with strong relevance for simulation and educational applications.

Second, we introduce **Guided Decoding**, a simple, training-free logit-fusion method that Reviewer **oYbY** highlights as practical, lightweight, and broadly applicable.

Finally, as emphasized by Reviewers **CaA5** and **wkof**, our paper provides **meaningful quantitative metrics**, PDS and SDS that offer the first structured evaluation framework for this problem. The experimental results are convincing and demonstrate clear improvements over existing prompting-based baselines.

## **Improvements During Discussion Period**

During the discussion period, we carefully reviewed the shared concerns raised by the reviewers and made targeted clarifications and revisions to strengthen the work.

First, we addressed the concern raised by Reviewers **wkof** and **oYbY** regarding the **quantitative evaluation of reasoning quality (Property 3)**. We conducted an additional analysis on 100 GSM8K samples using an LLM-as-a-judge framework to categorize reasoning errors. The results show that Guided Decoding influences not only final answer correctness but also the internal structure of reasoning: higher proficiency levels exhibit progressively fewer logical inconsistencies, heuristic shortcuts, and incorrect assumptions. The full experimental details and results are now provided in **Appendix F**.

Second, following Reviewer **oYbY**’s suggestion, we expanded our explanation of the **expert default bias**. We clarify that our conclusion is supported by both **systematic quantitative** evidence and illustrative **qualitative examples**. Section **3.1** has been revised for improved clarity and alignment with reviewer feedback.

Third, to address Reviewer **oYbY**’s concern regarding **model size effects**, we conducted new experiments using larger backbone models, including **Mistral 24B** and **Qwen 32B**. Our findings show that scaling to larger models does *not* cause few-shot prompting to surpass Guided Decoding. Across both model families, Guided Decoding maintains consistently lower PDS/SDS scores and provides the most stable proficiency control. Updated results are reported in **Appendix G**.

Fourth, in response to Reviewer **wkof** and **CaA5**’s request for **additional ablation on the number of guidance prompts**, we performed further experiments using 4 and 5 prompts. The findings reinforce that 3 prompts offer a strong balance between controllability and efficiency. While 2 prompts already outperform prompting baselines, increasing to 4 or 5 prompts yields only marginal improvements. These extended results are available in **Appendix H**.

Fifth, we addressed the **scalability concern** raised by Reviewer **CaA5** by providing a complete set of guidance prompts in **Appendix J**, along with a detailed explanation of how they were constructed and how the design process can be adapted to other domains in rebuttal.

Finally, several remaining concerns stemmed from **misunderstandings or insufficiently explanations**, such as notation clarity and the framing of guidance prompts, noted by Reviewers **wkof** and **oYbY**. We have clarified these points directly in our responses and refined the manuscript to reduce potential ambiguity for future readers.

In summary, through detailed clarifications, additional experiments, and expanded analyses, we believe we have addressed all concerns raised by the reviewers and substantially improved the quality and clarity of the paper. We sincerely thank the reviewers for their thoughtful feedback and constructive suggestions, which have greatly strengthened this work.

Best Regards,

Authors

---

### Meta-Review · Area_Chair_AmSU · 2026-01-02

**Summary:**

This paper proposes Guided Decoding, a decoding-time method to control the reasoning proficiency of LLMs by interpolating token-level logits from an expert reference prompt and one or more guidance prompts that encode flawed reasoning. The paper frames this as addressing an "expert default bias" in LLM role-play, where models remain too competent even when instructed to simulate average or low-proficiency users. The authors introduce new metrics to evaluate smoothness and alignment of proficiency control and present experiments across multiple QA benchmarks and model backbones. Reviewers generally agreed that the problem setting is well-motivated and that the method is simple and practical. However, concerns were raised about the level of technical novelty relative to prior decoding methods, the reliance on hand- or model-generated guidance prompts, and the strength of the underlying modeling assumptions used for evaluation.

A central concern was that the proposed decoding approach is closely related to existing guided or contrastive decoding methods, with limited technical novelty beyond applying these ideas to the setting of proficiency control. Although the framing around "expert default bias" is intuitive, reviewers were unconvinced that the method introduces a fundamentally new decoding mechanism rather than an incremental adaptation of prior work. Reviewers also expressed reservations about the reliance on curated or model-generated guidance prompts that encode specific reasoning errors. Finally, the evaluation relies on newly proposed metrics that assume a roughly linear relationship between proficiency level and task performance. Although this assumption is motivated, reviewers viewed it as a strong modeling choice that underpins many of the quantitative results and may not hold universally across tasks or settings.

**Reviewer Concerns:**

The rebuttal partially addressed several reviewer concerns:
- The authors added additional experiments and analyses (e.g. reasoning-quality diagnostics, more backbone models, and further ablations on the number of guidance prompts), which helped clarify the behavior of the method and strengthened the empirical section.
- Clarifications around the expert default bias and notation issues improved the presentation and resolved some misunderstandings raised by reviewers.

However, several significant concerns remain:
- The technical contribution of the decoding method itself remains incremental and closely related to existing contrastive or guided decoding approaches. While the application to proficiency control is novel, the underlying mechanism is viewed by reviewers as a modest extension rather than a fundamentally new technique.
- The approach relies on carefully constructed guidance prompts that encode specific error patterns. Although the authors argue this is scalable and low-bias, reviewers remained unconvinced that this dependency is fully addressed or that the method generalizes robustly beyond the tested settings.
- The proposed evaluation metrics (PDS/SDS) depend on a linear proficiency-performance assumption that, while motivated, is still a strong modeling choice and may not universally reflect how reasoning proficiency manifests across tasks.

Taken together, these issues limit the perceived impact and generality of the contribution.

**Reviewer Scores:**

- wkof would likely keep their 6.
- CaA5 would likely keep their 4, though it's possible that through more discussion they'd flip to a 6.
- oYbY is unlikely to change scores from 2 due to their fundamental objections to the framing, novelty, and evaluation assumptions that were not fully alleviated.

This paper was truly borderline for me.

---

### Decision · Program_Chairs · 2026-01-26

Reject